# VEGF-Related Germinal Polymorphisms May Identify a Subgroup of Breast Cancer Patients with Favorable Outcome under Bevacizumab-Based Therapy—A Message from COMET, a French Unicancer Multicentric Study

**DOI:** 10.3390/ph13110414

**Published:** 2020-11-23

**Authors:** Jocelyn Gal, Gérard Milano, Patrick Brest, Nathalie Ebran, Julia Gilhodes, Laurence Llorca, Coraline Dubot, Gilles Romieu, Isabelle Desmoulins, Etienne Brain, Anthony Goncalves, Jean-Marc Ferrero, Paul-Henri Cottu, Marc Debled, Olivier Tredan, Emmanuel Chamorey, Marco Carlo Merlano, Jérôme Lemonnier, Marie-Christine Etienne-Grimaldi, Jean-Yves Pierga

**Affiliations:** 1Epidemiology and Biostatistics Department, Centre Antoine Lacassagne, University Côte d’Azur, 06100 Nice, France; jocelyn.gal@nice.unicancer.fr (J.G.); emmanuel.chamorey@nice.unicancer.fr (E.C.); 2Cancer Pharmacogenetics and Radiogenetics Unit (UPRC) 7497, Centre Antoine Lacassagne, University Côte d’Azur, 33 avenue de Valombrose, 06100 Nice, France; 3Scientific Research National Center (CNRS), Inserm, Ircan, FHU-Oncoage, Centre Antoine Lacassagne, University Côte d’Azur, 06100 Nice, France; brest@unice.fr; 4Oncopharmacology Unit, Centre Antoine Lacassagne, University Côte d’Azur, 06100 Nice, France; nathalie.ebran@nice.unicancer.fr (N.E.); Laurence.LLORCA@nice.unicancer.fr (L.L.); Marie-Christine.ETIENNE-GRIMALDI@nice.unicancer.fr (M.-C.E.-G.); 5Department of Biostatistics, Institut Claudius Regaud, IUCT Oncopole, 31300 Toulouse, France; Gilhodes.Julia@iuct-oncopole.fr; 6Medical Oncology Department, Institut Curie, St Cloud, 92210 Paris, France; coraline.dubot@curie.fr (C.D.); etienne.brain@curie.fr (E.B.); paul.cottu@curie.fr (P.-H.C.); jean-yves.pierga@curie.fr (J.-Y.P.); 7Medical Oncology Department, Centre Val d’Aurelle-Paul Lamarque, 34298 Montpellier, France; gilles.romieu@icm.unicancer.fr; 8Medical Oncology Department, Centre Georges François Leclerc, 2100 Dijon, France; IDesmoulins@cgfl.fr; 9Medical Oncology Department, Institut Paoli-Calmettes, 13900 Marseille, France; GONCALVESA@ipc.unicancer.fr; 10Medical Oncology Department, Centre Antoine Lacassagne, University Côte d’Azur, 06100 Nice, France; jean-marc.ferrero@nice.unicancer.fr; 11Medical Oncology Department, Institut Bergonie, 33000 Bordeaux, France; m.debled@bordeaux.unicancer.fr; 12Medical Oncology Department, Centre Leon Berard, 69008 Lyon, France; olivier.tredan@lyon.unicancer.fr; 13Oncology Department, S. Croce & Carle Teaching Hospital, 12100 Cuneo, Italy; mcmerlano@tiscali.it; 14Research & Development Departement, Unicancer, 94270 Paris, France; j-lemonnier@unicancer.fr; 15Department of Medical Oncology, Paris University, 70006 Paris, France

**Keywords:** breast cancer, precision medicine, pharmacogenetics, bevacizumab, SNP, VEGF, overall survival

## Abstract

The prospective multicenter COMET trial followed a cohort of 306 consecutive metastatic breast cancer patients receiving bevacizumab and paclitaxel as first-line chemotherapy. This study was intended to identify and validate reliable biomarkers to better predict bevacizumab treatment outcomes and allow for a more personalized use of this antiangiogenic agent. To that end, we aimed to establish risk scores for survival prognosis dichotomization based on classic clinico-pathological criteria combined or not with single nucleotide polymorphisms (SNPs). The genomic DNA of 306 patients was extracted and a panel of 13 SNPs, covering seven genes previously documented to be potentially involved in drug response, were analyzed by means of high-throughput genotyping. In receiver operating characteristic (ROC) analyses, the hazard model based on a triple-negative cancer phenotype variable, combined with specific SNPs in *VEGFA* (rs833061), *VEGFR1* (rs9582036) and *VEGFR2* (rs1870377), had the highest predictive value. The overall survival hazard ratio of patients assigned to the poor prognosis group based on this model was 3.21 (95% CI (2.33–4.42); *p* < 0.001). We propose that combining this pharmacogenetic approach with classical clinico-pathological characteristics could markedly improve clinical decision-making for breast cancer patients receiving bevacizumab-based therapy.

## 1. Introduction

The vascular endothelial-derived growth factor A (VEGFA) is a positive regulator of angiogenesis and a coordinator of vascular homeostasis. In addition, the capacity of VEGFA to act as an immunosuppressor is well-established [1]. Therefore, targeting VEGFA and/or its receptors (VEGFR1, 2 and 3) constitutes a direct means to influence the tumoral environment. Bevacizumab, a VEGFA targeting monoclonal antibody, entered clinical practice more than 15 years ago [2] and is currently the most prescribed antiangiogenic treatment. Indeed, bevacizumab is indicated for colorectal cancer, breast cancer, non-small cell lung carcinoma (NSCLC), glioblastoma, renal cell carcinoma, ovarian cancer and cervical cancer [3]. The immunosuppressive properties of VEGFA have recently opened new perspectives regarding bevacizumab’s immunomodulatory potential. Indeed, bevacizumab has recently demonstrated clinical benefits for NSCLC and hepatocellular carcinoma in combined cancer immunotherapy [2,4]. However, despite intense investigations, reliable biomarker signatures that would allow us to pinpoint the individuals most susceptible to benefit from bevacizumab-based treatments remain to be identified and validated. Germline genetic polymorphisms may influence the outcome of cancer treatments in several aspects, such as individual variability in drug metabolism, immune response [5] and target availability [6,7].

The French multicentric cohort study COMET aimed to identify biological factors that could predict the clinical benefit of a bevacizumab–paclitaxel combination therapy as first-line treatment in metastatic breast cancer. The COMET study was specifically designed to survey key candidate genes potentially linked to the pharmacogenetics of both bevacizumab and paclitaxel. Notably, the elimination of paclitaxel is known to be influenced by *CYP2C8* and *ABCB1* polymorphisms [8,9]. In addition, by following a small cohort of metastatic breast cancer patients receiving another bevacizumab-based combination (bevacizumab–taxane), we previously established the prognostic value of the single nucleotide polymorphism (SNP) 936 C > T (rs3025039) in *VEGFA* [10]. The ECOG 2100 trial demonstrated the importance of polymorphisms in the *VEGFA* and *VEGFR2* genes for predicting the outcome of patients treated with the bevacizumab–paclitaxel combination compared with paclitaxel alone [11].

In the present study, 13 different SNPs across six genes were analyzed in an effort to identify a specific SNP signature significantly associated with bevacizumab-paclitaxel treatment outcome in terms of survival among metastatic breast cancer patients. The risk score analyses evaluating their predictive value as stand-alone variables versus when combined with common pathophysiologic criteria used in clinical practice are presented.

## 2. Results

### 2.1. Study Population and Clinical Characteristics of Tumors

Out of the 342 patients included, 306 genomic DNA samples were available for gene polymorphism analysis. The patients’ disposition is presented in Figure 1. The patients’ characteristics, the disease and the treatment baselines are summarized in Table 1. The mean age at inclusion was 55 years (range: 28–80). At initial diagnosis, the main histological type was invasive ductal carcinoma (79.5%). Most patients had received prior neoadjuvant/adjuvant chemotherapy (*n* = 210, 68%). Patients presented a small number of metastatic sites (90% had less than three metastatic sites); 62.5% of patients had a metastasis-free interval greater than 24 months. One hundred and thirty-five patients (48.5%) had histological Grade II tumors and 43.5% had Grade III tumors. The vast majority of patients had a positive hormone receptor status (*n* = 206, 76.5%) and 23.5% had triple-negative subtype tumors. Twenty-eight patients (9.35%) showed a complete clinical response (CR), 139 (46.5%) a partial response (PR), 101 (33.8%) a stable disease (SD) and 131 (10.35%) disease progression (PD).

### 2.2. Association between Progression-Free Survival and Clinico-Pathological Features and SNPs

The median follow-up was 50 months (95% CI (47–54)). The 5-year overall survival (OS) was 21% (16–28) and the 5-year progression-free survival (PFS) was 4% (2–7). The median OS and PFS were equal to 32 months (95% CI (28.5–36)) and 11 months (95% CI (9.5–11.5)), respectively.

The relationship between PFS and patient characteristics and 13 SNPs was assessed by univariate analyses (Appendix A). PFS was significantly associated (Table 2 and Appendix A) with histological Grade III (HR = 1.6, 95% CI (1.2–2.0); *p* < 0.001), triple-negative subtype (HR = 1.9, 95% CI (1.4–2.5); *p* < 0.001), *VEGFA* rs699947 A/A (HR = 1.3, 95% CI (1–1.8); *p* = 0.042), *VEGFA* rs833061 C/C (HR = 1.40, 95% CI (1.0–1.8); *p* = 0.028), *VEGFA* rs2010963 G/C or C/C (HR = 0.77, 95% CI (0.61–0.97); *p* = 0.026) and *VEGFR1* rs9582036 C/A or C/C (HR = 1.4, 95% CI (1.10–1.70); *p* = 0.01). As shown in Appendix A, patients with the triple-negative subtype and histological Grade III tumors had a median PFS of 5.9 months (95% CI (4.50–7.85)), while patients with hormonal receptor-positive and histological Grade I or II tumors had a median PFS of 12.94 months (95% CI (11.83–14.82); *p* < 0.001).

Through multivariable analyses, three distinct predictive models (Model A: genetic; Model B: clinico-pathological; Model C: combined) were evaluated for PFS (Table 3). Two significant variables remained predictive of a poor prognosis: *VEGFA* rs833061 C/C (HR = 1.35, 95% CI (1.02–1.80); *p* = 0.032) and *VEGFR1* rs9582036 C/A or C/C (HR = 1.34, 95% CI (1.06–1.71); *p* = 0.011) for Model A, and histological Grade III (HR = 1.5, 95% CI (1.15–1.95); *p* < 0.001) and triple-negative subtype (HR = 1.75, 95% CI (1.29–2.37); *p* < 0.001) for Model B. For Model C, histological Grade III (HR = 2.0, 95% CI (1.50–2.65); *p* < 0.001), triple-negative subtype (HR = 2.26, 95% CI (1.64–3.12); *p* < 0.001) and *VEGFA* rs833061 C/C (HR = 1.39, 95% CI (1.01–1.91); *p* = 0.037) were associated with a shorter PFS. As shown in Figure 2A, the AUC for the risk score of Model C at 24 months was higher compared with Models A and B (AUC_model C_: 74.75 (95% CI (67.98–81.51)); AUC_model B_: 63.12 (95% CI (55.22–71.03)); AUC_model A_: 60.97 (95% CI (53.08–68.87)). Dichotomization risk score was then carried out using the receiver operating characteristic (ROC) curve at 24 months (optimal cut-off value: −0.22) separating the poor from good prognosis groups of patients with a sensitivity of 58% and a specificity of 87%. As shown in Figure 3A, patients assigned to the poor prognosis group (*N* = 155) presented a median PFS of 8.71 months (95% CI (7.46–9.92)) and patients in the good prognosis group (*N* = 93) presented a median PFS of 13.53 months (95% CI (11.70–15.90); *p* < 0.001). The Cox regression analysis indicated that patients assigned to the poor prognosis group were significantly more prone to disease progression (HR = 2.05, CI 95% (1.55–2.72); *p* < 0.001).

### 2.3. Associations among Overall Survival, SNPs and Patient Characteristics

Univariate analyses of OS were performed (Appendix A). Among all tested variables, OS was significantly correlated (Table 3 and Appendix A) with histological grade III (HR = 1.4, 95% CI (1.1–1.9); *p* = 0.011), triple-negative subtype (HR = 2.4, 95% CI (1.7–3.3); *p* < 0.001), *VEGFA* rs699947 A/A (HR = 1.5, 95% CI (1.1–2.1); *p* = 0.007), *VEGFA* rs833061 C/C (HR = 1.6, 95% CI (1.1–2.1); *p* = 0.005), *VEGFA* rs1870377 T/T (HR = 1.4, 95% CI (1–1.8); *p* = 0.021) and *VEGFR1* rs9582036 C/A or C/C (HR = 1.4, 95% CI (1.1–1.9); *p* = 0.01). As shown in Appendix A, triple-negative subtype and hormonal receptor-positive status were predictive of the worst and best prognosis, respectively. More precisely, triple-negative subtype patients had a median OS of 14.4 months (95% CI (12–21.7)) when having a histological Grade III tumor and a median OS of 15.7 months (95% CI (11.6–NA)) with histological Grade I or II tumors. In contrast, hormonal receptor-positive patients had a median OS of 38 months (95% CI (33.9–45.6)) when having a histological Grade I or II tumor and a median OS of 31.8 months (95% CI (25.3–41.7)) for histological Grade III patients.

As for PFS, three distinct predictive models (i.e., Model A: genetic; Model B: clinico-pathological; Model C: combined) were evaluated (Table 4). Three significant variables remained predictive of poor prognosis for Model A: *VEGFA* rs833061 C/C (HR = 1.59, 95% CI (1.15–2.20); *p* = 0.003), *VEGFR1* rs9582036 C/A or C/C (HR = 1.34, 95% CI (1.01–1.78); *p* = 0.034) and *VEGFR2* rs1870377 T/A or A/A (HR = 1.39, 95% CI (1.04–1.86); *p* = 0.02). Only the histological Grade III variable remained predictive for Model B (HR = 3.42, 95% CI (2.44–4.81); *p* < 0.001). With regard to Model C, the triple-negative subtype (HR = 4.22, 95% CI (2.96–6.01); *p* < 0.001), *VEGFA* rs833061 C/C (HR = 1.40, 95% CI (1.00–1.98); *p* = 0.049), *VEGFR1* rs9582036 C/A or C/C (HR = 1.56, 95% CI (1.15–2.10); *p* = 0.003) and *VEGFR2* rs1870377 T/A or A/A (HR = 1.69, 95% CI (1.24–2.3); *p* < 0.001) were associated with shorter OS. As shown in Figure 2B, at 60 months, the AUC for the risk score of Model C was better than the ones from Models A or B (AUC_model C_: 70.23 (95% CI (57.84–82.62)); AUC_model B_: 62.99 (95% CI (59.94–66.05)); AUC_model A_: 49.53 (95% CI (38.43–60.63))). At 60 months and based on Model C, the optimal cut-off value was 0.06 and risk groups were associated with a sensitivity of 33% and a specificity of 100%, respectively.

As shown in Figure 3B and as defined by Model C, patients falling into the good prognosis group had a longer median OS (15.2 months; 95% CI (12.3–19.2)) compared with patients assigned to the poor prognosis group (37.1 months; 95% CI (32.6–41.2); *p* <0.001) (Figure 3B). Notably, patients assigned to the poor prognosis group had a death hazard ratio of 3.21 (95% CI (2.33–4.42); *p* < 0.001).

## 3. Discussion

The present pharmacogenetic study was based on a homogeneous group of 306 patients with respect to cancer type and advanced disease status with an identical treatment protocol. An optimal predictive model for OS was obtained with the tumor subtype (triple-negative) and polymorphisms within the gene encoding the target of bevacizumab itself: *VEGFA* (rs833061 C/C), as well as within the genes encoding its receptors (*VEGFR1*; rs9582036 C/A or C/C and *VEGFR2*; rs1870377 T/T). The germline genome is sometimes considered to have limited applications regarding cancer therapy optimization [7]. However, the present study supports the idea that gene polymorphisms may help predict bevacizumab-based treatment outcomes. Bevacizumab is increasingly used outside its classical clinical area of prescription [12]. Moreover, the immunomodulatory properties of bevacizumab have recently led to very promising associations in combined immunotherapies for NSCLC and hepatocellular carcinomas [2]. Thus, the present data may further encourage the identification and validation of reliable and much needed bevacizumab biomarkers. The clinical impact of *VEGFA* gene polymorphisms was recently examined in patients with advanced colorectal cancer treated with bevacizumab in association with chemotherapy. So far, the prognostic value of *VEGFA* rs699947 is unclear [13,14]. In contrast, *VEGFA* rs833061 demonstrated strong prognostic value in advanced colorectal cancer patients and, to a lesser extent, in breast cancer patients [13]. Notably, our data further support the prognostic value of *VEGFA* rs833061 for breast cancer patients.

Considering that rs833061 is located in the 5’-UTR of *VEGFA*, we then sought to investigate whether this specific polymorphism might have an impact on *VEGFA* expression, especially considering that this SNP is associated with nine SNPs (r^2^ > 0.8) all located in the promoter region or in the first introns of *VEGFA*. Together, these SNPs were associated with the high expression of *VEGFA* in thyroid tissue or different alternative splicing events in the left heart ventricle vs. muscle (according to GTEX portal query results). These observations warrant further investigation to establish whether this SNP is functionally linked to the phenotype (poor outcomes) observed in the context of the present study.

rs9582036 is located on the *FLT1* gene (also known as *VEGFR1*) and is part of a cluster of 15 SNPs with close linkage disequilibrium (0.6 < r^2^ < 0.8). Fourteen of these SNPs are intronic, including the synonymous variant rs9582036 associated with an alternative splicing event in adipose tissue and the thyroid (according to the GTEX portal). All SNPs are located between the 25th and the 28th intron (30 exons in total). An in silico analysis did not show evidence of reliable variations. Nevertheless, the presence of a shorter isoform (ENST0000061 7835.4) that starts at Exon 28 and codes for the last 85 amino acids of the protein could be regulated differently due to the surrounding SNPs. Further phenotypic analyses would be required to understand the relationship between the SNP and the isoform (ENST00000617835.4). The rs1870377 T/A variant located on the *KDR* gene (also known as *VEGFR2*) introduces a missense glutamine (Gln) to histidine (His) substitution at the amino acid position 472, potentially impacting VEGFR2 degradation.

The present study contains a certain number of limitations, as we aimed to focus on a predefined set of SNPs. GWAS would be required in order to gain a broader insight into the SNPs involved in bevacizumab-based treatment outcomes. However, GWAS necessitates a larger number of patients. The impact in terms of OS conferred by the specific SNPs touching the targets of bevacizumab and its cellular receptors merits further investigation in the case where bevacizumab is associated with immunotherapy. In this respect, recent experimental data [15] highlighted a direct effect of VEGFA on endothelial receptors that modifies T cell diapedesis and allows TRegs rather than T cytotoxic cells to migrate to the tumor bed, thus decreasing the tumor’s immune capacity. The SNPs described herein that have an impact on the outcome of bevacizumab-based treatment may thus be of potential prognostic value in the context of bevacizumab–checkpoint inhibitor combinations, a particularly promising domain of application for this widely used antiangiogenic and immunomodulatory agent.

## 4. Materials and Methods

### 4.1. Study Design and Patients

This prospective, multicenter COMET trial covered a cohort of 342 consecutive patients included between 2012 and 2015. The inclusion criteria were: age > 18 years, histologically confirmed metastatic HER2- breast cancer, ECOG ≤ 2, life expectancy ≥ 12 weeks, patients receiving first-line chemotherapy with bevacizumab (10 mg/kg every 2 weeks) and paclitaxel (90 mg/m^2^ at Day 1, Day 8 and Day 15). Treatment was repeated every 4 weeks, according to routine practice, until disease progression or unacceptable toxicity. The exclusion criteria were prior chemotherapy for metastatic breast cancer, concomitant endocrine therapy or radiation therapy with curative intent for oligometastatic disease. The tumor response was defined according to Response Evaluation Criteria in Solid Tumours (RECIST 1.1) criteria as complete response (CR), partial response (PR), stable disease (SD) or progressive disease (PD). The objective response rate (ORR) was defined as the proportion of patients whose best overall response over the entire treatment period was CR or PR. Informed written consent was obtained from each patient. The French Institutional Ethics Committee approved the study in June 2012 and it was registered (NCT01745757).

### 4.2. SNP Selection and Genotyping

Genomic DNA was extracted from blood samples using the commercially available Maxwell^®^ 16 LEV Blood DNA Kit (#AS1290, Promega^©^, Madison, WI, USA). High-throughput genotyping of germline DNA was performed by MassARRAY [16] (AGENA Bioscience^®^, San Diego, CA, USA) using the custom panel of 18 SNPs covering 7 objectively preselected genes corresponding to the drug response: *VEGFA* (rs833061, rs2010963, rs3025039, rs1570360, rs699947), *VEGFR1* (rs9582036), *VEGFR2* (rs2071559, rs2305948, rs1870377), *IL8* (rs4073), *CYP2C8* (rs11572080, rs10509681), CYP450 (rs2740574, rs776746) and *ABCB1* (rs2229109, rs2032582, rs1045642, rs2032582). Independent controls were performed by conducting SNPs’ analyses either at another center (rs1045642 and 1128503) or by a different technique (RFLP: rs833061, rs2010963, rs1570360). For 18 SNPs, the minor allele frequency was ≥5% in Caucasians, according to SNPpedia (http://www.snppedia.com) and the Ensemble database (http://www.Ensembl.org).

Five SNPs were excluded from analyses, 4 SNPs (*VEGFA* rs1570360; *ABCB1* rs2032582; CYP450 rs2740574, rs776746) for not respecting Hardy–Weinberg equilibrium and 1 SNP (*CYP2C8* rs10509681) for pairwise linkage disequilibrium (LD) higher than 0.8 (*CYP2C8* rs11572080) [17]. Overall, the analyses were performed on 13 SNPs in 6 genes (Appendix A).

### 4.3. In Silico Analysis

The potential functional impact of SNPs was investigated using several in silico applications. HaploReg (http://www.broadinstitute.org/mammals/haploreg/haploreg.php) was used to collect all linkage disequilibrium and to examine the haplotype blocks within the genes. Regulome DB (http://www.regulomedb.org/) was accessed to explore the chromatin status, conservation and regulatory motif alterations within sets of genetically-linked variants. The GTEX Portal (https://gtexportal.org/home/) served to identify all *cis*-eQTL SNPs affecting the expression of genes of interest and as a microRNA binding-site prediction tool. SNPMIR (https://www.genomique.info/joomla/) was used to predict whether a SNP within the 3’-UTR of the genes of interest would disrupt/eliminate or enhance/create a microRNA binding site.

### 4.4. Statistical Analysis

An evaluation of the missing data rate was performed on 13 SNPs. All SNPs included in the analysis had a missing data rate of less than 10%. Missing genotypes were assigned using multiple imputations by chained equations (MICE) [18]. Dominant and recessive models were investigated to test possible associations between SNPs and progression-free survival (PFS) and overall survival (OS). For each SNP, risk alleles were coded as 1 and non-risk alleles as 0. For each SNP, the hazard ratio (HR) and 95% confidence intervals were calculated by Cox regression for associations between the genotype and PFS or OS. PFS was defined as the time between inclusion and the date of disease progression or death due to any cause. OS was defined as the time between inclusion and death due to any cause. Patients showing no event (death or progression) or lost to follow-up were censored at the date of last contact. PFS and OS were performed using the Kaplan–Meier method. Median follow-up with a 95% confidence interval was calculated with the reverse Kaplan–Meier method. For PFS and OS, we constructed a genetic and a clinico-pathological predictive model using a multivariable Cox regression method with backward elimination (*p*-value ≤ 0.05). The variables selected in each of the previous two predictive models were included in a third multivariable Cox regression with backward elimination to create a combined predictive model. For each model, the proportional hazard assumption was checked using statistical tests and graphical diagnostics based on the scaled Schoenfeld residuals [19]. For each multivariable model, a classifier predicting the risk of progression or death was based on the linear predictor given by the model. The predictive ability of each model was compared using the area under the ROC curve (AUC). For PFS and OS, the clinical endpoint was fixed at *t* = 24 months and *t* = 60 months, respectively. Risk groups for predictive models were obtained using the best threshold value to obtain the highest sensitivity with a specificity of at least 80%. The sensitivity was defined as the proportion of patients experiencing progression or death before the time *t* attributed to the poor prognosis group. The specificity was defined as the proportion of event-free patients beyond the time *t* attributed to the good prognosis group. A *p*-value of <0.05 was considered statistically significant and all tests were two-sided. All statistical analyses were performed with R.3.5.2 software on Windows^®^ and the survMisc [20], timeROC [21], Survminer [22] and LDcorSV packages [23].

## 5. Conclusions

This study identified genetic polymorphisms related to a bevacizumab–paclitaxel treatment combination and established a genetic risk score in association with the clinico-pathological characteristics of patients. The highlighted risk groups identified patients with a poor or good prognosis and could be used to improve clinical decision-making.

## Figures and Tables

**Figure 1 pharmaceuticals-13-00414-f001:**
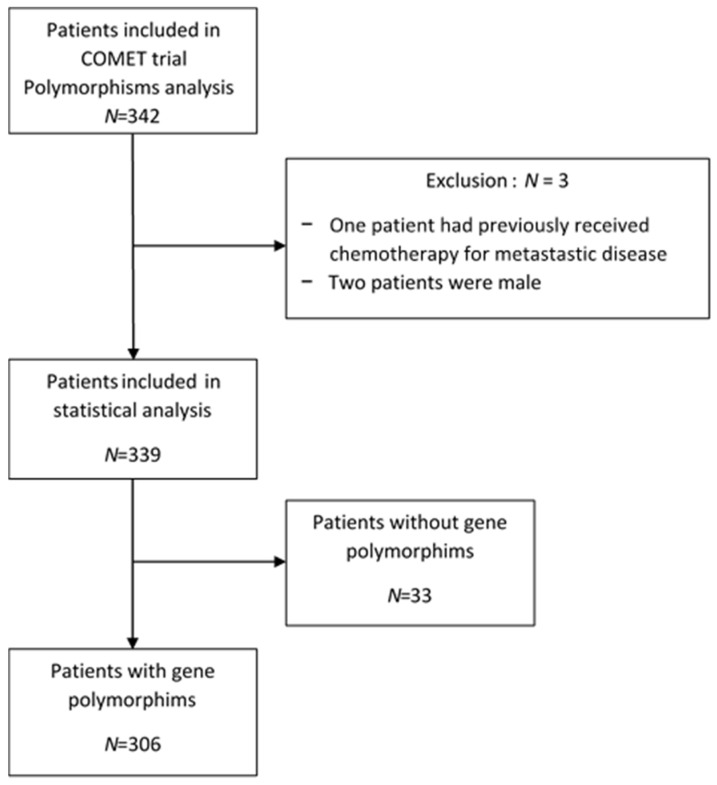
Flow diagram of the study participants.

**Figure 2 pharmaceuticals-13-00414-f002:**
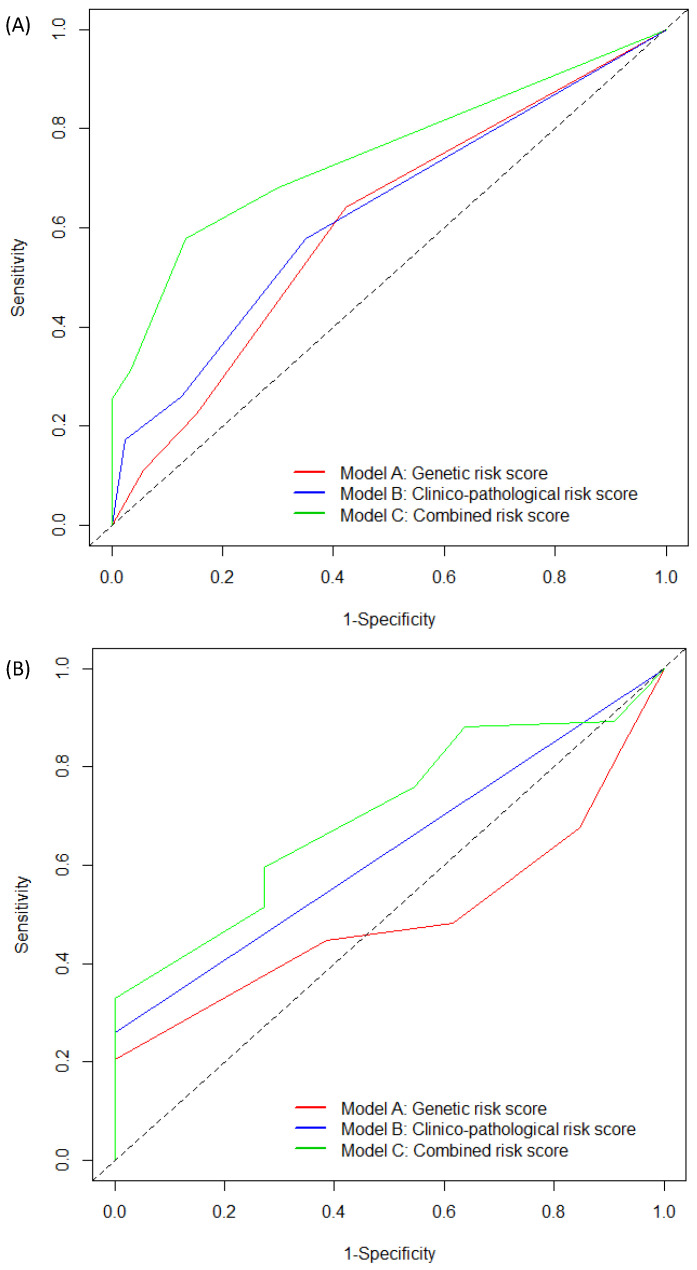
Receiver operating characteristic (ROC) curves and AUC for risk scores for three models. (**A**) PFS at 24 months; (**B**) OS at 60 months.

**Figure 3 pharmaceuticals-13-00414-f003:**
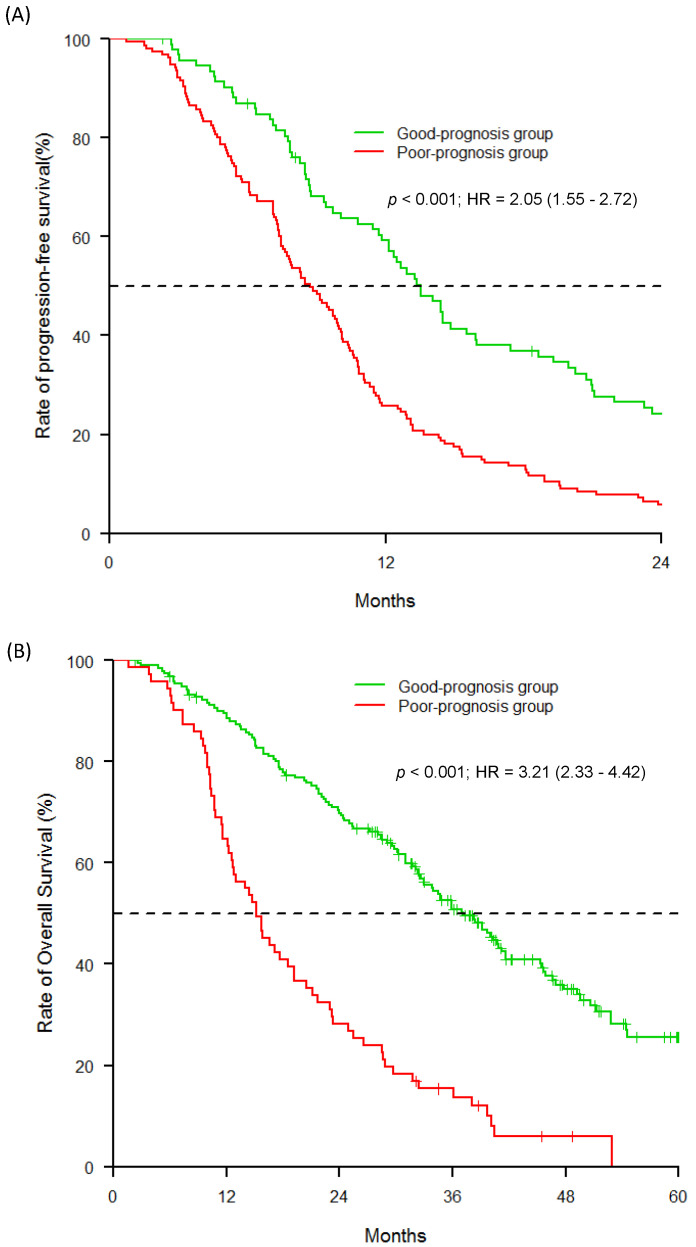
PFS and OS of the patients within different risk groups based on the predictive model. (**A**) PFS; (**B**) OS.

**Table 1 pharmaceuticals-13-00414-t001:** Patients and tumor characteristics at baseline.

Characteristics	Number of Patients (*N* = 306)	%
Mean age _(min–max) at inclusion_	55.5_28–80_	
Histology		
Invasive ductal carcinoma	237	79.5
Invasive lobular carcinoma	38	12.5
Mixed (ductal and lobular)	10	3.5
Other	13	4.5
Menopausal status at inclusion		
Premenopausal	88	29
Postmenopausal	213	71
Unknown	5	
Performance status at inclusion		
0	172	56
1/2	134	43
Histological grade		
I	23	
II	135	8
III	121	48.5
Unknown	27	43.5
Subtype		
Triple-negative	64	
Hormonal receptor-positive	206	23.5
Unknown	36	76.5
Tumor stage at initial diagnosis		
pT_0_/pT_1_	109	
pT_2_	84	46.5
pT_3_/pT_4_	41	36
Unknown	72	17.5
Axillary lymph node status at initial diagnosis		
pN0	83	
pN+	153	35
Unknown	68	65
Prior (neo)adjuvant chemotherapy		
Yes	210	68
No	96	32
Number of metastatic sites		
<3	211	
≥3	24	90
Unknown	71	10
Metastasis-free interval		
0 months	17	7
0–24 months	71	30.5
24 months	147	62.5
Unknown	71	
Metastatic sites		
Liver	97	41
Distant node	60	25.5
Bone	37	15.5
Skin	13	5.5
Lung	19	8.0
Other ^1^	9	4.5
Unknown	71	

^1^: Soft tissue, central nervous system, breast, peritoneum, glands.

**Table 2 pharmaceuticals-13-00414-t002:** Univariate analysis for the polymorphisms and clinical characteristics according to progression-free survival (PFS) and overall survival (OS).

		**Progression-Free Survival**		**Overall Survival**	
**Significant Gene Polymorphisms**		**HR 95% CI**	***p*-Value**	**HR 95% CI**	***p*-Value**
**VEGFA**	rs699947				
	C/C or A/C	1	Referent	1	-
	A/A	1.3 (1–1.8)	0.042	1.5 (1.1–2.1)	0.007
	rs833061				
	T/T or T/C	1	Referent	1	-
	C/C	1.4 (1–1.8)	0.028	1.6 (1.1–2.1)	0.005
	rs2010963				
	G/G	1	Referent	-	-
	G/C or C/C	0.77 (0.6–0.97)	0.026	-	-
**VEGFR1**	rs9582036				
	A/A	1	Referent	1	-
	C/A or C/C	1.4 (1.1–1.7)	0.010	1.4 (1.1–1.9)	0.010
**VEGFR2**	rs1870377				
	T/A or A/A	-	-	1	-
	T/T	-	-	1.4 (1–1.8)	0.021
**Significant Clinical Characteristics**		**HR 95% CI**	***p*-Value**	**HR 95% CI**	***p*-Value**
**Histological grade**					
	Grade I or II	1	Referent	1	-
	Grade III	1.6 (1.2–2)	<0.001	1.4 (1.1–1.9)	0.011
**Subtype**					
	Hormonal receptor-positive	1	Referent	1	-
	Triple-negative	1.9 (1.4–2.5)	<0.001	2.4 (1.7–3.3)	<0.001

**Table 3 pharmaceuticals-13-00414-t003:** Multivariate analysis according to PFS.

Significant Gene Polymorphisms		Progression-Free Survival					
		Model A: Genetic		Model B: Clinico-Pathological		Model C: Combined	
		HR [95% CI]	*p*-Value	HR [95% CI]	*p*-Value	HR [95% CI]	*p*-Value
***VEGFA***	rs833061						
	T/T or T/C	1	-	-	-	1	-
	C/C	1.35 (1.02–1.80)	0.032	-	-	1.39 (1.01–1.91)	0.037
***VEGFR1***	rs9582036						
	A/A	1	-	-	-	-	-
	C/A or C/C	1.34 (1.06–1.71)	0.011	-	-	-	-
**Significant clinical characteristics**							
Histological grade	Grade I or II	-	-	1	-	1	-
	Grade III	-	-	1.50 (1.15–1.95)	<0.001	2.0 (1.50–2.65)	<0.001
Subtype	Hormonal receptor-positive	-	-	1	-	1	-
	Triple-negative	-	-	1.75 (1.29–2.37)	<0.001	2.26 (1.64–3.12)	<0.001

**Table 4 pharmaceuticals-13-00414-t004:** Multivariate analysis according to OS.

Significant Gene Polymorphisms		Overall Survival					
		Model A: Genetic		Model B: Clinico-Pathological		Model C: Combined	
		HR [95% CI]	*p*-Value	HR [95% CI]	*p*-Value	HR [95% CI]	*p*-Value
***VEGFA***	rs833061						
	T/T or T/C	1	-	-	-	1	-
	C/C	1.59 (1.15–2.20)	0.003	-	-	1.40 (1.00–1.98)	0.049
***VEGFR1***	rs9582036						
	A/A	1	-	-	-	1	-
	C/A or C/C	1.34 (1.01–1.78)	0.034	-	-	1.56 (1.15–2.10)	0.003
***VEGFR2***	rs1870377						
	T/T	1	-	-	-	1	-
	T/A or A/A	1.39 (1.04–1.86)	0.020	-	-	1.69 (1.24–2.30)	<0.001
**Significant clinical characteristics**							
Histological grade	Grade I or II	-	-	-	-	-	-
	Grade III	-	-	-	-	-	-
Subtype	Hormonal receptor-positive	-	-	1	-	1	-
	Triple-negative	-	-	3.42 (2.44–4.81)	<0.001	4.22 (2.96–6.01)	<0.001

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
