# Peer review of "VEGF-Related Germinal Polymorphisms May Identify a Subgroup of Breast Cancer Patients with Favorable Outcome under Bevacizumab-Based Therapy—A Message from COMET, a French Unicancer Multicentric Study"

_pharmaceuticals, 2020, doi:10.3390/ph13110414_

Round 1

Reviewer 1 Report

This is a well designed study.

My only question is did you looked inflammation markers before and after treatment? 

What is the reason you did not apply GWAS or whole genome sequencing in this study?

Author Response

  • Responses to reviewer 1:

This is a well-designed study.

1. My only question is did you looked inflammation markers before and after treatment? 

Response: Although some experimental data suggest that VEGF may inhibit adhesion formation through inflammatory responses it was not the purpose of present work to incorporate markers of inflammation.

2. What is the reason you did not apply GWAS or whole genome sequencing in this study?

Response: GWAS was not applied in this study due to:

  1. The strategy of a supervised approach covering given genetic markers
  2. The relatively limited number of subjects at our disposal contraindicating GWAS which necessities a large number of cases

Reviewer 2 Report

Figure 1 legend: Please expand a bit more on what this figure shows. The legend "flow chart" is not enough.

Some inconsistencies in terminology notes. Please consider using the term "germline" DNA throughout the whole text rather than germinal.

Please define a bit better how your group of 306 individuals are homogeneous. The fact that the have the same type of cancer and received same treatment is not sufficient. How did you take into account other confounding factors? Describe their similarities but also their differences a bit better. How did you take into account for example whether or not these patients were taking additional therapeutic drugs for other conditions they may have? 

Author Response

  • Responses to reviewer 2:
  1. Figure 1 legend: Please expand a bit more on what this figure shows. The legend "flow chart" is not enough.

Response: The legend of figure 1 (flow chart) has been modified by “Flow diagram of the study participants”. We developed this point on line 84-85.

  1. Some inconsistencies in terminology notes. Please consider using the term "germline" DNA throughout the whole text rather than germinal.

Response: We replaced the term “germinal” by the term "germline".

  • Please define a bit better how your group of 306 individuals are homogeneous. The fact that they have the same type of cancer and received same treatment is not sufficient. How did you take into account other confounding factors? Describe their similarities but also their differences a bit better. How did you take into account for example whether or not these patients were taking additional therapeutic drugs for other conditions they may have? 

Response: Multicenter COMET trial is a prospective trial with precise inclusion and exclusion criteria (see lines 195-201). All patients were treated identically by first-line chemotherapy with bevacizumab (10 mg/kg every 2 weeks) and paclitaxel (90 mg/m2 at day 1, day 8, day 15).

Moreover, for each multivariable model the proportional hazards assumption was checked using statistical tests and graphical diagnostics based on the scaled Schoenfeld residuals (lines 248-249). The objective of these statistical analyzes was that patients included in the multivariate analysis were made as homogeneous as possible in final.